# Understanding Farmers' Adoption of Sustainable Agriculture Innovations: A Systematic Literature Review

**José Rosário** [1,2,*], **Lívia Madureira** [1], **Carlos Marques** [3] and **Rui Silva** [1]

1 Centre for Transdisciplinary Development Studies (CETRAD), University of Trás-os-Montes e Alto Douro (UTAD), Quinta de Prados, 5000-801 Vila Real, Portugal
2 Center for Advanced Studies in Management and Economics, University of Évora, 7000-809 Evora, Portugal
3 Mediterranean Institute for Agriculture, Environment and Development (MED), University of Évora, Apartado 94, 7006-554 Evora, Portugal
* Correspondence: josevictorino1@gmail.com

**Abstract:** Adoption of sustainable agriculture innovations is acknowledged to be an effective response to agro-ecological challenges, such as climate change, pests, drought, natural catastrophes, and food insecurity. However, its level of dissemination is still low across the world, particularly in the Global South. There is a need for a better understanding of the adoption determinants of these innovations in order to enhance them. This paper presents a systematic literature review focused on the use of sociopsychological determinants to understand the adoption of sustainable agriculture innovations, combining conventional bibliometric analysis with the method of vote-count. This method enabled an evaluation of the ability of the determinants considered by the models, as well as respective sociopsychological constructs, to explain the innovation adoption. Our results show a significant growth in the research employing theory and models built on sociopsychological factors to understand the decision-making processes undertaken by farmers in the context of the adoption of sustainable agriculture innovations. The development of statistical models and techniques, such as the structural equation model (SEM), has facilitated the inclusion of a growing set of sociopsychological variables. However, our review highlights that the selection of the sociopsychological constructs used by research to explain farmers' adoption of sustainability innovations relies mainly on constructs defined for other decisional contexts, such as the adoption of innovations by firms in other sectors. Hence, the low ability evidenced by the models to explain farmers' adoption behavior is due to a poor selection of constructs. The review highlights that this poor selection is a result of repetition of constructs, such as attitude, subjective norms, and little inclusion of other relevant constructs such as knowledge. The paper suggests the need for a better selection of the innovation determinants and measurement of respective constructs adjusted to the case of agriculture and the specificities of the diverse geographical farming contexts.

**Keywords:** sustainable agriculture; innovation adoption; behavioural models; SLR; TPB; TAM

## 1. Introduction

Sustainable agriculture innovations are acknowledged to tackle challenges faced by agriculture such as climate change, pest, drought, natural catastrophes, and food insecurity [1–3]. However, due to the low level of adoption of these innovations, there is a need to investigate the determinants of adoption through models [1]. Models can provide a representation of complex relationships between variables to explain adoption [1].

Several studies [2–6] suggested the inclusion of sociopsychological constructs in the modeling of the farmer's decision to adopt or not adopt agricultural innovations. However, as far as we know, there are no studies to date that evaluate the way in which this integration of sociopsychological constructs in modeling has been carried out [1]. This paper addresses a research gap in the literature consisting of a poor selection of the determinants included

in the models explaining the adoption of sustainability agriculture innovations [1]. The paper combines conventional bibliometric analysis with the method of vote-count in order to address this research gap. Despite the vast literature available to understand the practice of modeling the adoption of innovations in agriculture, there are several reasons to conduct this review. The main motivation is the fact that, although there are several literature reviews [1,3,4,6,7] on this topic, none of them focused on evaluating the selection of sociopsychological constructs in the context of the adoption of sustainability agriculture innovations. This research gap justifies the need for more studies in order to understand how academics can deliver a clear message [1] to policymakers so that the adoption of sustainability agriculture innovations can be reinforced. This is because the introduction of these agricultural innovations has met only partial success [8], and the adoption rate remain low mainly in Global South [9]. Thus, this paper focuses on a critical analysis of the available evidence respecting the most frequent models and variables used by studies addressing the adoption of sustainability agricultural innovations. The paper comprises a systematic literature review using the bibliometric analysis method combined with the vote-count method to answer the research questions, as well as to offer insights into future research on this topic.

Regarding the topic of sustainability agriculture innovations, adoption research is needed to investigate the discrepancies of variable definition in the context of the conceptual models and how they have been used in empirical studies. In this respect, the contribution of Meijer et al. [10] emphasized the role of extrinsic and intrinsic factors for the uptake of agricultural innovations in the decision-making process, arguing that intrinsic factors such as the knowledge, perceptions, and attitudes of the potential adopter toward innovation play a key role. Several studies using different conceptual models have been carried out to understand the factors that influence the adoption of a diversity of sustainability agriculture innovations such as agroforestry systems [10], fertilizer tree systems [11], conservation agriculture [12], organic farming [13,14], green pesticides [15], and other agricultural practices that contribute to sustainability.

Decision-making models and analytical methods on the topic of agricultural innovation adoption emerged from Ryan [16], which explained the diffusion of agricultural innovation using a mathematical formulation. Fifteen years later, Griliches [17] introduced regression modeling and demonstrated that the innovation decision-making process and its acceptance could be explained through economic analysis. Rogers [18] established the diffusion of innovations (DOI) theory that proposes the diffusion and the adoption of innovative technologies, to be determined by the compatibility and complexity of the new technology, the prospective end-user's characteristics, the person's perception, and knowledge regarding the technology and the type of communication channels identified as mass, group, and individual. In addition to the specific theories of innovation in agriculture, some models, such as the theory of reasoned action (TRA) and the theory of planned behavior (TPB) proposed by Ajzen [19], were imported from psychology into the agriculture field in order to better understand the sociopsychological aspects of farmers' adoption. According to Burton [20] decomposed TPB seems to be more encompassing, and all these theories have often been applied intensively in empirical studies relating to consumer behavior, manufacturing industries, advertising campaigns, information technologies, and software sciences [10,21–24].

Nevertheless, its use in agriculture is still limited. Foguesatto et al. [3] reviewed available empirical evidence analyzing the factors positively or negatively influencing the adoption of sustainable agriculture practices and concluded that the inclusion of sociopsychological factors to model farmers' decision to adopt was scarce. However, this study focused on UM, TPB, and TRA and did not include other sociopsychological models, likely explaining why the constructs were poorly measured in the studies reviewed by the authors. Borges et al. [4] analyzed the variables influencing farmers to adopt agriculture innovations. According to this study, the results of the research employing the utility maximization (UM) theoretical framework showed explanatory variables to

have an insignificant effect on the innovation adoption, whereas the TPB results showed that correlations between sociopsychological constructs were quite positive, but poorly measured. These previous review studies were focused mainly on TPB and UM theory, leaving out other adoption models. These approaches have not permitted a complete overview of the limitations of different models. However, all these studies had a significant contribute to the body of literature in this topic and were relevant to identify the research gap to be addressed in this review.

The basic concept of sustainability innovations corresponds to creating new or improved products, services, technologies, processes, and management techniques that produce environmental or social benefits and economic value, as argued by El Bilali [25]. Sustainable agriculture innovations can occur at the farmer's level, e.g., adopting green fertilizer, compost, conservation tillage, soil and water conservation, fallow, legume crop rotations, improved seed varieties, and use of animal manure, or outside the farm, e.g., adopting short distribution chains and cooperatives. This is defined as a process where sustainability concerns, such as environmental, social, and financial, are integrated into agricultural systems from idea generation through research and development (R&D) and commercialization, and then applied to products, services, and technologies, as well as new business and organization models [26].

This study contributes to the construction of relevant knowledge for policymakers to establish the design of policies more effective in promoting sustainable agriculture innovations. The main contribution of the review is to raise the awareness of researchers regarding the need to expand the dimensions to be represented by models analyzing the decision-making process concerning the adoption of sustainable agriculture innovations instead of focusing only on the dimensions already explored by the established models that tend to present an overlap in the constructs included. In the current study, it is argued that there is a need to include constructs such as efficacy, trust, awareness, and knowledge, given that these sociopsychological constructs have the potential to add value to the analysis of adoption modeling in the case of agriculture [27].

The remainder of this paper is structured as follows: Section 2 offers a literature review of the theoretical models used to explain agricultural innovation uptake. Section 3 describes the methodological approach. The results and discussion are provided in Section 4. Lastly, conclusions are presented in Section 5.

## 2. Literature Review on Agricultural Innovation Adoption Models

Models explaining farmers' decision-making process have been a significant research theme in adopting agricultural innovation [7]. "Innovation diffusion" is one of the well-known models, which states that the diffusion and the adoption of agricultural innovation, are determined by the compatibility and complexity of the new technology, the prospective end-user characteristics, the person's perception and knowledge regarding the technology and communication channels [18]. A vast amount of literature has demonstrated that UM and TPB are the theories that support the two main types of models most often used to analyze farmers' decisions to adopt an innovation. Nevertheless, there is a huge diversity of models, as described below. According to the TPB, the intention is the person's motivation to exert effort and enact the behavior. Attitude is the positive or negative evaluation of the performance of a particular behavior [19]. The models explain perceived behavior control as an individual's perceived ease or difficulty with a particular behavior performance. Social norms are the social pressure exerted on an individual to engage in a particular behavior. Ajzen's [19] TPB was criticized for neglecting emotional factors that may determine the intention of behavior while implicating that intention and perceived behavior control were better predictors of self-reported behavior than observed behavior [28]. However, it is widely used in research areas outside of psychology [1]. TPB has evolved from reasoned action theory, which states that human action depends on an individual's intention influenced by individual attitudes and social norms [4]. According to Adnan et al. [29], the main theories explaining farmers' decision-making are DOI and the theory of reasoned

action (TRA). The TRA states that human behavior considers available information and their actions' advantages and disadvantages. Furthermore, the action or behavior to be performed is immediately determined by the person's intention. The intention is determined by subjective norm and attitude toward the behavior, both mediated by the relative importance of attitudinal and normative components [30]. Indeed, TPB, the motivational model, social cognitive theory, the technology acceptance model (TAM), and the values–beliefs–norms (VBN) are associated with the decision-making process. TAM is a model applied to forecasting the use of technologies on the basis of knowledge and attitudes toward that technology. The model was adapted from TRA [19] and introduced the concepts of perceived usefulness, understood as the degree to which someone believes that using a particular system would enhance their job performance and perceived ease of use, defined as the degree to which a person believes that using a particular system would be free of effort, as argued by Davis et al. [31]. Both beliefs are used to explain the intention to use of a technology. VBN theory explains a wide range of pro-environmental behaviors such as environmental activism, curb recycling, and sustainable resource management, focusing on the role that these personal norms play as a moral obligation for the accomplishment of a specific action or abstaining from the behavior [32]. The theory provides a framework where the pro-environmental behavior results from an interactive combination of variables such as values, beliefs, awareness of the consequences, ascription of responsibility, and personal norms [33].

The UM theory assumes that farmers make rational choices to maximize their profit and wellbeing within their resource capabilities [34,35]. Additionally, this theory is an economic model relying on the logic that farmers can adopt agricultural innovations to maximize expected utility by comparing two situations: expected utility from adopting the innovation ($U_i^1(\prod)$) and from not adopting it ($U_i^0(\prod)$). The decision to adopt only occurs if the net expected utility exceeds zero ($U_i^1(\prod)$) > ($U_i^0(\prod)$), where the expected maximization utility function is U(.) = MaxU($\prod$) [36]. In this equation, U(.) is the expected utility that depends on vector constraints ($\prod$), namely, the farmers' resources, wealth, and specific characteristics. This model is widely used; however, today, the consensus is the consideration of sociopsychological factors, rather than relying only on economic models, as well as including other sources of information, its effects, and the characteristics of the agricultural practices themselves [22] to capture the full picture of the adoption process. For instance, Blazy et al. [37] evaluated the willingness to adopt agroecological innovations to reduce pesticide use for banana production in the West Indies as an ex-ante evaluation, employing a multi-attribute choice model (choice experiment). Jara-Rojas et al. [36] used logit models to estimate the probability of adoption for particular technologies or techniques and revealed that social and natural capital increases the likelihood of adoption. The main limitation of models that econometrically determine the importance of various explanatory variables is that they only explain the decision to adopt or not adopt and do not explain the intensity of adoption [7]. Economic constraint theory demonstrates how the distribution of resources such as land, capital, labor, liquidity, and other inputs explains the use and scaling up of new technological innovations [22].

According to Prochaska et al. [38], the transtheoretical model (TTM), also known as the stage of change (SOC) model, provides a useful framework for understanding how individuals intentionally change their behaviors, with or without professional intervention. The TTM defines change as a gradual, continuous, and dynamic process. It holds that individuals do not go directly from old behaviors to new behaviors but progress through a sequence of stages: pre-contemplation, contemplation, preparation, action, and maintenance. The classification criteria of these models are provided in Table 1. Recently, the contribution of Sutherland et al. [39] to the literature was the development of the triggering change model (TCM), stating that a trigger event leads to an active assessment stage followed by an implementation and a consolidation stage.

**Table 1.** Models explaining the agricultural innovation uptake decision process and its classification criteria (adapted with permission from Munguia et al. [1]).

| Models Classification Criteria | Description | Dimensions | Examples | Knowledge and Subject Area | Source |
|---|---|---|---|---|---|
| Nature of the concept | Focused on adopter's characteristics | Individual decision-making process models | Bass-like models, | Marketing | Bass [40] |
| | | | Information flow models | | Lindner et al. [41] |
| | | | Dynamic risk/economic model | Economics | Abadi Ghandim and Pannell [42] |
| | | | Utility maximization model | Economics | Rahm M. Huffman [43] |
| | | Extended behavioral models | TPB | Psychology | Ajzen [19] |
| | | | Adopt model | Multidisciplinary | Kuehne et al. [44] |
| | | | Goal-directed behavior | Psychology | Perugini and Bagozzi [45] |
| | Focused on technology | Individual decision-making process models | Task–technology fitness model | Information Systems | Goodhue and Thompson [46] |
| | | | Economic constraint theory | Economics | Aikens et al. [47] |
| | | Extended behavioral models | Technology acceptance model | Multidisciplinary | Davis and Venkatesh [48] |
| | | | Satisfaction models | Marketing | Meyer and Allen [49] |
| Analysis unit | Singular adopter | - | Social cognitive theory | Sociology | Venkatesh et al. [50] |
| | Population adopter | - | Diffusion models | Sociology | Rogers E. [18] |

## 3. Methods

A systematic literature review (SLR) was conducted by Mallet et al. [51]. The SLR was built on a strategy outlined to strengthen the quality of the SLR, preventing eventual bias and difficulties in its replication. A protocol was established to define the search methods, screening procedures, data extraction, and evaluation [52]. The search was conducted in the online databases WoS (Web of Science) and Scopus, the collected data were downloaded in Bibtex format, and the R Studio software was used to eliminate duplicates and to create a unified database. The software R Bibliometrix 3.0 was used to conduct the bibliometric analysis [53]. Figure 1 synthesizes the steps followed to implement the protocol defined for the SLR presented by this paper.

The first step consisted of conducting the search in the online databases for the selected keywords and the keyword combinations. The option for a broader search entailed the choice of the keyword combinations presented in Table 2. A total of 1069 records were extracted (578 from WoS and 491 from Scopus) [3]. In a second step, eligibility filters were applied, comprising (1) a chronological filter, i.e., the period between 2000 and 2021, (2) a document filter, selecting solely scientific articles published in English language, and (3) scientific major areas, including agricultural and biological sciences, business, management and accounting, economics, and social sciences. The application of these filters reduced the number of documents to a total of 1052. In the third step, the collected data were downloaded in Bibtex format, and the R Studio software (version 4, London, UK) was used to eliminate 186 duplicates and to create a unified database. The final database used for bibliometric analysis included 866 papers by 2672 authors, published in 282 dif-

ferent high-impact journals, as evidenced by the fourth step comprising the bibliometric analysis [54] conducted using the software R Bibliometrix 3.0 (Naples, Italy).

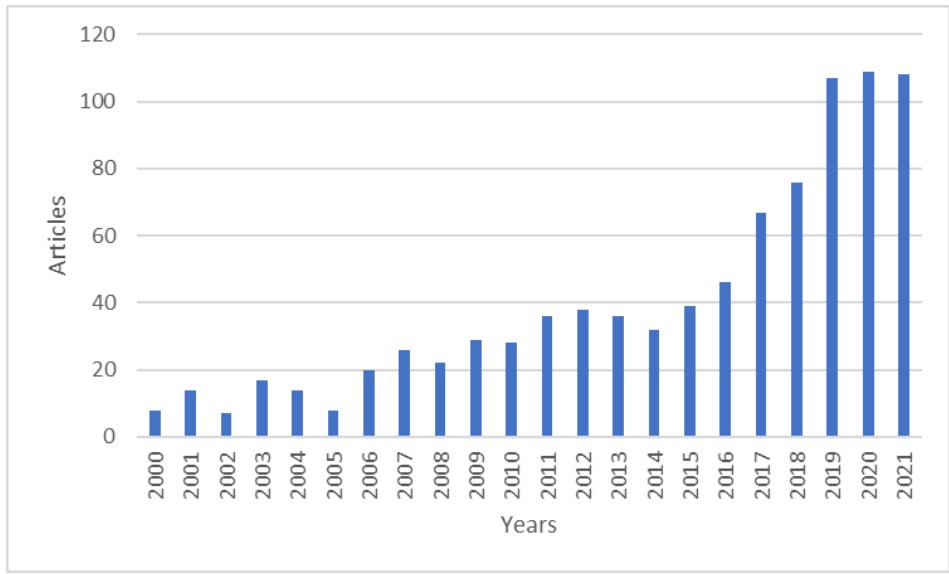

**Figure 1.** Publications per year (2000–2021). Note: The figure was generated using R Studio software (Bibliometrix tool) with the bibliographic coupling method [53], analyzing data from WoS and Scopus, with the author being the unit of analysis.

**Table 2.** Description of the main steps to implement SRL.

| Phases | Procedure | Criteria | Output |
|--------|-----------|----------|--------|
| Step 1 | Search with key words in databases | "agricult* AND sustainable innovation AND adoption AND models OR theory OR adopt OR transtheoretical AND model OR bass-like AND model OR step-hazard OR diffusion OR goal-directed AND behavior AND attitude OR tpb OR task-technology AND fit AND model OR technology AND acceptance AND model OR desire OR intention OR adoption AND behavior" | n = 1069 (578 WoS + 491 SCOPUS) |
| Step 2 | Automatic screening with filters | Scientific Field: agricultural and biological sciences, business, management and accounting, economics, and social sciences Time period: 2000–2021 Type of document: scientific articles in English | n = 1052 documents |
| Step 3 | Construction of unified database | 186 duplicate documents were removed using R Studio. A unified database was created. | n = 866 documents |
| Step 4 | Bibliometric analysis | Bibliometric analysis with R Studio. Data analysis, visualization, and interpretation | n = 866 documents |
| Step 5 | Selection of articles from database created in step 3 | At least one of the models investigated the adoption of sustainable agricultural innovation adoption as a dependent variable | n = 62 documents |
| Step 6 | Vote-count of construct and models | Independent variables: constructs of models with positive or negative effect on adoption with critical level of 10% significant | n = 62 documents |

A bibliometric analysis depicts the main trends in a research topic, describes the development and structure of a research domain, and synthesizes elements that help to define the structure of the knowledge and its systematization. In addition, it provides opportunities for researchers and journals to set up a research agenda on the research topic [55]. Regarding the importance of bibliometric analysis in terms of science mapping, Zupic and Carter [54] stated that criteria such as who (related to authors), what (keywords indicating main topics), where (locations), when (year of publication), and with whom (collaborative networks) were first identified by science mapping techniques [54,56].

In addition to the broad SRL performed through bibliometric analysis, an in-depth review was conducted with two additional steps. A fifth step consisting of a manual search in the final database resulted from the third step of the protocol, as described above. Two criteria needed to be simultaneously fulfilled for the document inclusion in the second database: (1) the authors applied a theoretic model belonging to the family of behavior sociopsychological models and its extensions; (2) dependent variables of the model(s) included the adoption of sustainable agriculture innovations. In the final step (sixth step), the vote-count method was employed to identify the frequency and effect of constructs included in the models. The vote-count was adopted as it consists of an approach that adds to just narrative reviews by including tables of significance counts [3,6]. A variable was considered to have a significant effect on the adoption (dependent variable) if the estimated parameter was at the critical level of 10% significance [4]. In order to get a clear understanding, the analysis comprised a diversity of sociopsychological models and determinants rather than relying on the commonly employed TPB. These sociopsychological models encompass TPB, TAM, TRA, TTM, motivation opportunity ability (MOA), protection motivation theory (PMT), VBN, social network analysis (SNA), technology–organization–environment (TOE) [57], unified theory of acceptance and usage of technology (AUT2) moral obligation model [55,58], diffusion of innovations theory, Venkatesh's model [50], Marcus model [59], social capital analysis [60], multiplicity model [59], social cognitive theory [61], and situational factor model [62]. We follow the same criteria presented by Borges et al. [4] in order to understand if expanding the sample can influence the results in terms of determinants of adoption behavior within the scope of TPB, and other behavioral models emerging in response to criticism and narrowing the gap of previous models.

The presentation of the results, in the next section, starts by presenting the main results of the bibliometric review of the large database extracted automatically, with 866 documents. The second part presents an in-depth systematic review and the vote-count analysis of the studies (62 documents extract from the larger database) that used sociopsychological models to approach the decision-making process in adopting or not adopting sustainable agricultural innovations.

## 4. Results and Discussion

### 4.1. Bibliometric Analysis

The results of the bibliometric analysis distinguished two publication periods: a first period from 2000 to 2009 with an oscillating level of publication, and a second period from 2010 to 2021 where the annual number of publications increased steeply, with 2015 as a turning point, which can be explained by the launching of the SDGs, as illustrated in Figure 1. The steep increase in publications in the second period addressing the adoption of sustainable agricultural innovation is likely a consequence of a shift in the research agenda driven by the food prices crisis in 2008, which returned agriculture to global agendas [63].

However, regarding the theoretical–conceptual frameworks in the area of agricultural innovation, two main shifts occurred outside of the period considered by our SLR. Firstly, a shift in theoretical perspective from the diffusion of innovation/transfer of technology (central perspective in the 1960s), through farming systems research (in the 1970s and 1980s), and then agricultural knowledge and information systems (in the 1990s), and finally to agricultural innovation systems from 2000 to the present [64].

The bibliometric analysis conducted using R Studio software (Bibliometrix tool) with the bibliographic coupling method [53] allows relating the most relevant topics, countries, and collaboration networks between authors, as described in Figure 2. In this figure, a larger size of the colored rectangles indicates a higher frequency of a certain topic, keyword, or author within the collaboration network. The thickness of lines connecting authors, keywords, and countries depends on the number of connections. The three-field plots depict topics such as "sustainable agriculture", "adoption", "theory of planned behavior", "conservation agriculture", and "sustainability", which mostly gathered scholars from countries such as the USA, Malaysia, Germany, Kenya, and France. Furthermore, according

to Figure 3, research from the USA has focused on topics including sustainable agriculture, extension, and adoption. Research from China has investigated ecological intensification, planned behavior theory, and climate change. Nigerian researchers have focused on topics such as sustainable agriculture and farmers. Research from Germany has mostly investigated organic farming, sustainable development, and technology adoption. Spanish researchers have mostly investigated agroecology and cooperatives.

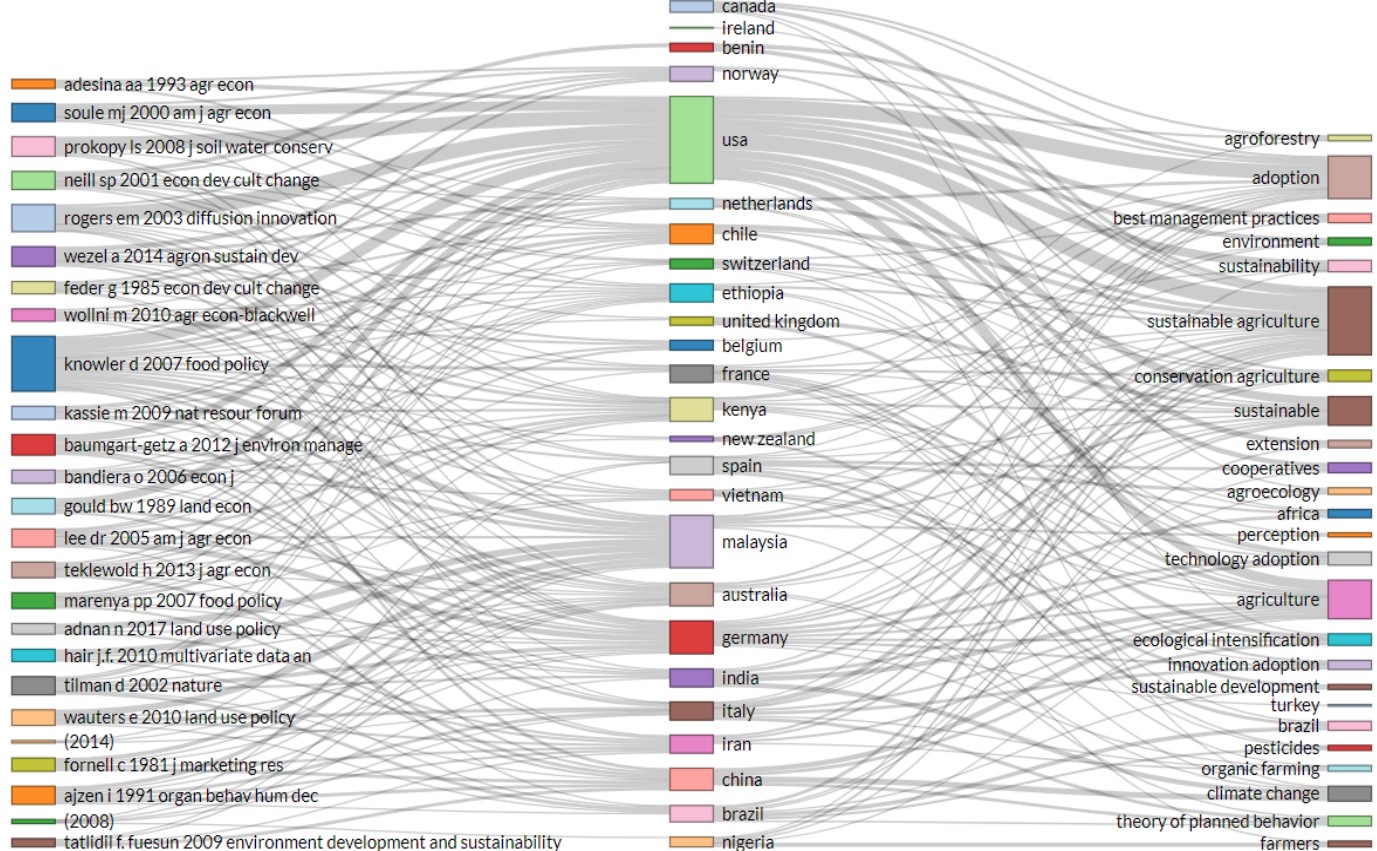

**Figure 2.** Three-field plots depicting the top 25 authors, countries, and keywords. Note: The figure was generated using R Studio software (Bibliometrix tool) with the bibliographic coupling method [53] analyzing data from WoS and Scopus.

Furthermore, Figure 2 shows that the subject under analysis, i.e., the models that explain adoption and its determinants, engaged researchers from all geographies, although there was greater production of knowledge on the topic in the Global North. Sustainable agriculture is presented in the figure as a central topic in which decision models for adopting sustainable innovations are framed. It is possible to observe a strong research activity on this subject in the USA, Malaysia, and other countries through the contribution of authors such as Knowler and Bradshaw [6], by reviewing existing research on economic profitability and other factors influencing adoption and providing a better understanding of on-farm adoption behavior through a classification of determinants with respect to farmer and farm household characteristics, comprising general farm characteristics, farm financial and managerial characteristics, exogenous determinants, attributes of innovation, and psychological determinants. Prokopy et al. [65] clarified adoption as an innovation decision process comprising at least five decision stages: knowledge, persuasion, implementation, and confirmation. Prior to their study, Feder et al. [7] highlighted at least two kinds of adoption: individual and aggregate. Individual adoption occurs at the farm level, whereas aggregate adoption occurs at the regional or national level, a process referred to as diffusion.

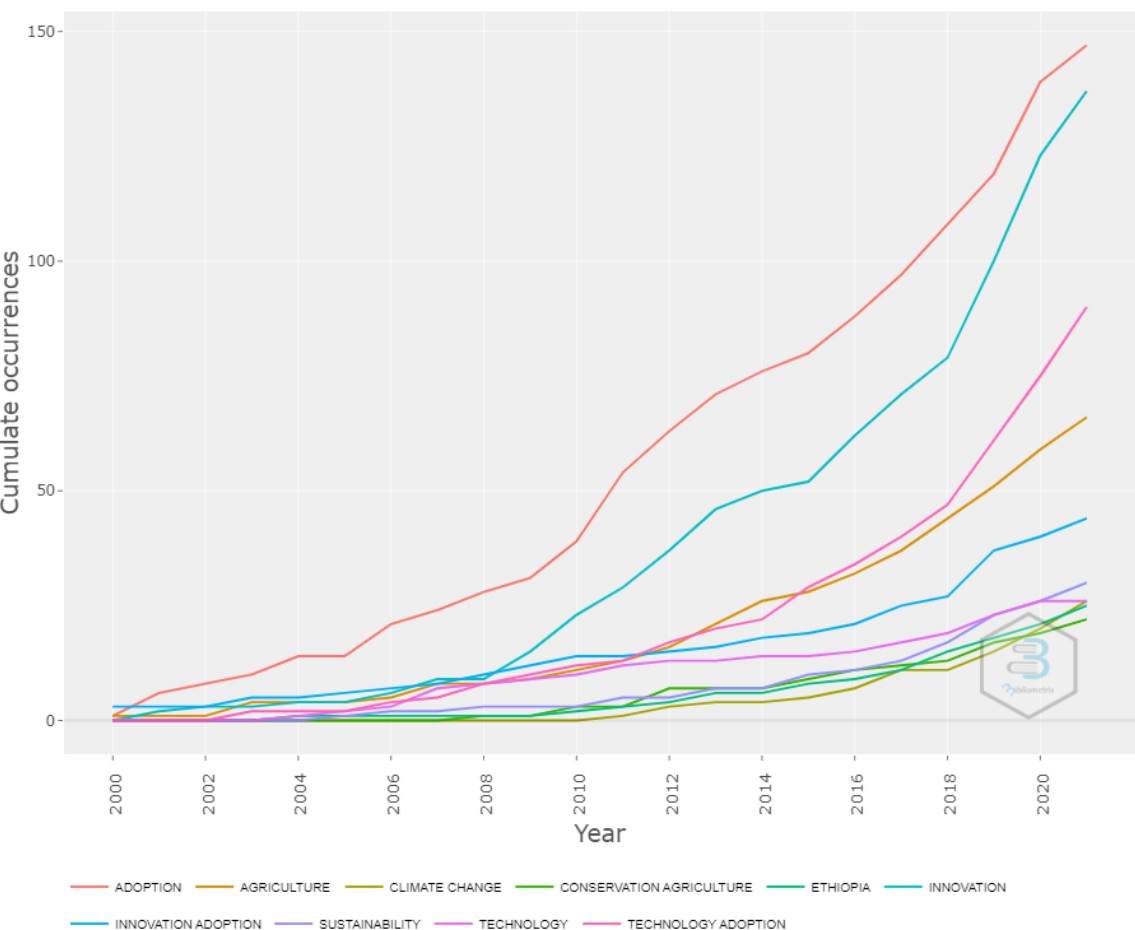

**Figure 3.** Dynamics of the conceptual structure growth in the period under review (2000–2020). Note: The figure was generated using R Studio software (Bibliometrix tool) with the bibliographic coupling method [53], analyzing keywords, i.e., terms obtained from the title, abstract, or document's body, previously extracted as data from WoS and Scopus online databases.

The production of knowledge about the adoption of agricultural innovations is globally concentrated in certain regions. The USA is the most productive country according to the number of citations. According to Haji et al. [66], the knowledge production of Anglo-American Societies (e.g., the USA, the UK, Canada, Australia, and New Zealand) is evidenced by a common trend in publications on management, social sciences, and decision science literature, accounting for about one-quarter of the articles in these research fields. It is important to highlight that Kenya and Ethiopia are the top representatives of knowledge production in the African context. Most of the research on this topic is about Africa by authors affiliated with European, Australian, and American institutions. European countries are at the forefront of adoption research, especially in the UK and the Netherlands, with Wageningen University at the top for publications.

The dynamics of the adoption according to type of agricultural innovations are illustrated in Figure 3. Through the conceptual structure growth, it is possible to perceive that words that grow the most are those to which researchers have paid the most attention. Its analysis is crucial to perceive the trend and evolution of sufficiently researched questions and those that still require further investigation. In the first period (2000 to 2009), conceptual growth was not evident, probably because of a temporary extension of the predominance of the technology transfer model [67] and utilitarian paradigm. The second period (2010 to 2021) presented a steep increase in research on concepts such as

"adoption" and "conservation agriculture". According to Naspetti et al. [68] the analysis of keyword growth is relevant since researchers can understand the dynamics of the conceptual structure, limit the scope of their studies, and delimitate the set of score documents to be included in the research.

The analysis of coauthor networks is depicted in the Figure 4. The figure was generated using R Studio software (Bibliometrix tool) with the bibliographic coupling method [53], analyzing the co-citation network in data from WoS and Scopus online databases for the period under analysis.

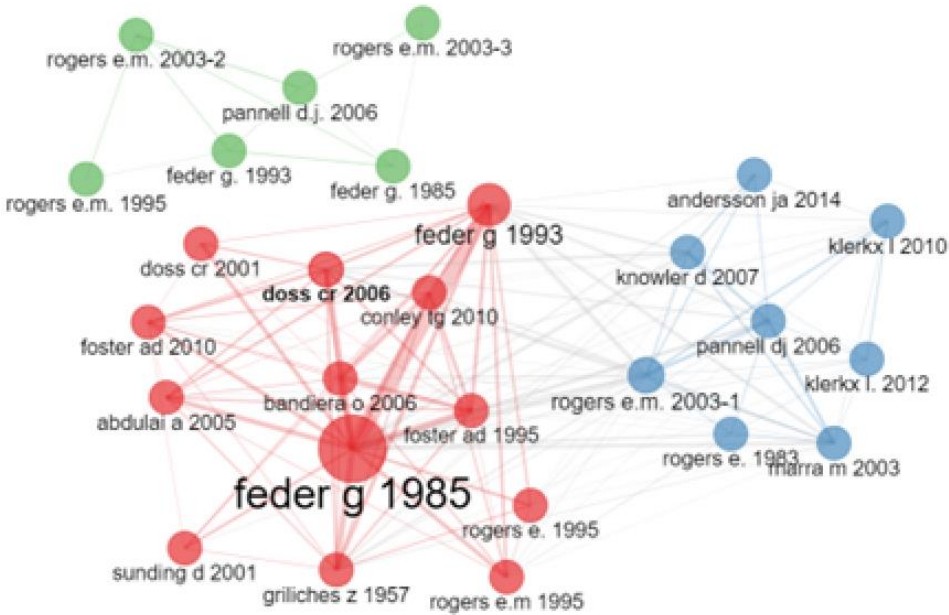

**Figure 4.** Co-citation networks. Note: The figure was generated using R studio software (Bibliometrix tool) with the bibliographic coupling method [53], analyzing the co-citation network in data from WoS and Scopus.

The three clusters encountered in the bibliometric review (Figure 5) can be differentiated by the connection, as well as the color and size of circles. The first cluster is agricultural extension, comprising several studies that addressed the adoption process by incorporating a systemic and multi-actor approach, as represented by Klerkx [69]. This cluster states that, since 1950, agricultural innovation approaches have evolved, adopting the transfer of technology orientation until 1980, before evolving to a systemic approach in recent decades with the adoption of the concept of AKIS (agricultural knowledge and innovation systems) that forms a broad governance framework for advisory services with other innovation support arrangements such as research, education, and innovation funding [70].

The second cluster is agricultural economics, represented by Feder, which analyzes the determinants of adoption relying on the utility maximization theory, suggesting economic factors as the main drivers of adoption. This cluster contribution is also evidenced by Feder et al. [7], who highlighted at least two kinds of adoption: individual and aggregate. Individual adoption occurs at the farm level, whereas aggregate adoption occurs at the regional or national level, a process referred to as diffusion. One of the first studies on aggregate adoption over time came from econometrics and was conducted by Griliches [17], who estimated the logistic function. The third cluster is focused on farmer behavior. This cluster's main contribution is the inclusion of sociopsychological factors as determinants of the adoption process. This cluster is represented by Rogers and is not interconnected with the other two, demonstrating the need for an integrated approach.

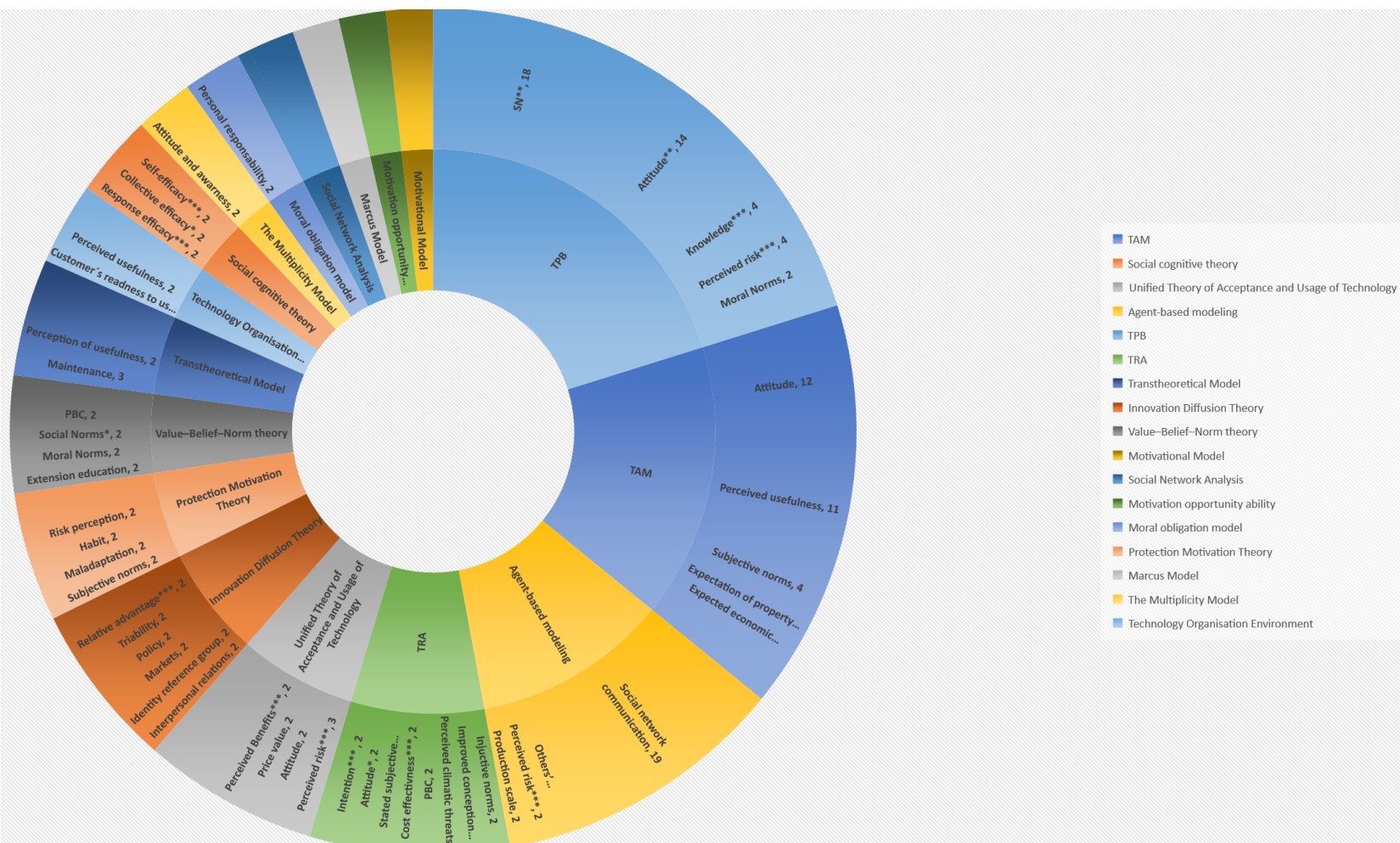

**Figure 5.** Sociopsychological models and respective constructs in the reviewed studies. Note: The level of the circle closest to the center includes the names of the models; in the next level, the constructs are represented, where the size of the area of different colors represents the frequency of those constructs in the reviewed studies, while the numbers in the figure are the vote-count results after analyzing the papers individually. *** $p < 0.01$, ** $p < 0.05$, * $p < 0.1$ indicate coefficients significant at respectively 1%, 5% and 10% levels related to the parametre of constructs in reviewed studies.

There are some authors represented in two clusters, such as Pannell and colleagues, who provided evidence on how the adoption research has innovatively recognized adoption decision making as a process. The authors highlighted that adoption is not an event or binary variable, while clarifying the adoption concept and suggesting many terms that are more specific than adoption. The authors also argued that research needs to include the collection of long-term datasets concerning adopting diverse practices and innovations, while understanding what determines the continuous and sustained adoption of conservation practices, why adoption varies between farms, and its practical implications.

Other authors emphasized the heterogeneity and strong multidisciplinary nature of adoption [8,71]. The contribution of these authors is relevant when designing future lines of investigation (Table 3) for modeling adoption. Additionally, the reformulation of public policy to include efforts to support women farmers' adoption of beneficial innovations in developing countries and the use of tools and approaches from marketing in public extension programs are proposed as mechanisms to achieve food security and the sustainable development goals (SDGs).

**Table 3.** Future lines of investigation as suggested by the research clusters.

| Proposed Future Lines of Research | Author | Cluster |
|---|---|---|
| - Modeling adoption decisions, not in dichotomous terms but considering the adoption intensity.<br>- Modeling the interrelation of innovations and policy. | Feder et al. [7] | Agricultural economics |
| - Evidence on policy effectiveness.<br>- Understanding whether innovations are adopted in the package, individually or in combination, following a sequence. | Feder et al. [8] | Agricultural economics |
| - Borrowing models from psychology and other social sciences to develop adoption models and other choices that recognize rationality.<br>- Research on the link among environmental regulation, research, development, and the adoption of new products. | Sunding and Zilberman [2] | Agricultural economics |
| - Using sampling approaches that allow the generalization of data from micro studies to higher levels of aggregation and adherence to standardized terms across studies. | Doss [72] | Agricultural economics |
| - More work needs to be conducted to clarify agricultural innovation systems conceptually and empirically. | Klerkx et al. [69] | Agricultural extension |
| - Researchers must be conscious of the type of practices that landholders adopt more readily and encourage a participatory process.<br>- Evaluating the complementarity between scientific and local knowledge, as well as multidisciplinary cooperation of researchers, to positively influence adoption. | Pannell et al. [73] | Agricultural extension |
| - Research on the farm-level adoption constraints that different types of farmers face, considering contextual factors focusing on the wider market, as well as the institutional and policy context. | Andersson and Souza [74] | Agricultural extension |
| - Gradual implementation of the proposed extension service in education, with a flexible approach to adapting the new system for greater effectiveness. | Rogers [18] | Farmer Behavior |
| - More research on the utility of social media and market recognition approaches (such as certificate schemes and consumer labeling) to influence best management practices adoption. | Liu et al. [55] | Farmer Behavior |

This analysis uncovered future lines of investigation that can constitute the research agenda in the coming decades for this topic. Gaps at the conceptual, methodological, and public policy levels have been pointed out by the authors. Thus, it is possible to group the three research axes suggested by the authors according to the cluster analysis. In

the agricultural economics cluster, the debate on the modeling of adoption is not closed; there is room to clarify the concept, as well as its modeling and intensity. Even from the methodological point of view, the debate is open on how to make it more feasible to import models from psychology and other sciences for integration with economics, as pointed out by Feder et al. [8]. In the agricultural extension cluster, there are research opportunities in the conceptual area of extension systems, as well as in the role of knowledge and its communication in the adoption of sustainable practices. Furthermore, researchers can investigate how the extension systems consider the circumstantial particularity of the agricultural household, as well as the issue of scale-up of the multi-actor approach [64]. In the farmer behavior approach cluster, the debate on the role and usefulness of social media in adoption, as well as the role of cultural context in adoption, remains open. Table 3 illustrates how an analysis of these lines of investigation could be carried out on this topic.

### 4.2. Vote-Count Analysis

This section presents the results of the in-depth review and vote-count analysis of the 62 scientific articles including sociopsychological model extensions enabling the evaluation of the explanatory power of the determinants considered by the authors. The evaluation was carried out using the vote-count method. This review refers solely to the to the research applying sociopsychological models. In the 62 studies reviewed, 18 different models were employed to analyze the sociopsychological determinants of innovation adoption. Twenty-two studies used TPB, nine used TAM, five used TRA, two used protection motivation theory (PMT), and two used diffusion of innovation theory. Iran was the country with the most studies, followed by China, the USA, and Germany. The TBP constructs with a significant and positive effect were attitude, followed by perceived behavior control (PBC) and social norms (SN) with a frequency of 19, 18, and 14, respectively, which is in line with the results of previous research. For instance, Zeweld et al. [22] combined decomposed theory of planned behavior with social cognitive theory, diffusion theory, and economic constraint theory. They found that attitudes and normative issues positively explain farmers' intentions to adopt minimum tillage and row planting. Furthermore, perceived control also positively affects the intention to apply minimum tillage. These authors highlighted that social capital and training positively affect subjective norms, which positively mediates the relationship among training, social capital, and intention. Applications of the TPB highlighted the attitudes, subjective norms, and perceived control as positively influencing farmers' intentions to adopt improved grassland management practices in Brazil [75]. The results also show that determinants from TAM, such as perceived usefulness (PU) and perceived ease of use (PEOU), have a positive and significant effect on adoption. Several studies confirmed this relationship in the agricultural field [76,77].

These results are comparable with the perspective of Meijer et al. [10]. They allow concluding that, with respect to the role of extrinsic and intrinsic factors in the uptake of innovation, considerable variation between studies exists and the impact of constructs is mixed, i.e., positive in some studies, but negative or non-significant in others. Although it seems to be difficult to establish the role of constructs in explaining the uptake of sustainable agricultural practices, it is possible to analyze which models and constructs were more employed by researchers (Figure 5) and understand why this happened. It is also important to underline that the measurement of the constructs included in the reviewed sociopsychological models was mainly conducted using a five-point Likert scale (1 = strongly disagree and 5 = strongly agree), while structural equation models (SEMs) were widely employed for statistical analyses.

The literature review revealed that researchers are not concerned with finding and applying perfect models, but with adapting the model selection to the research objectives or problems to be solved. Although there is no ideal model, there are models that are more suited to certain research topics or are more apt for the investigation of certain aspects due to the particularity of their constructs. For example, agent-based models are more focused on investigating aspects related to the evaluation of agricultural policies and the interaction

of producers, as well as their effect on the adoption process. On the other hand, researchers typically employed TAM or the unified theory of acceptance and use of technology (UTAUT) to study the determinants of intelligent and precision agriculture adoption.

On the other hand, statistical techniques have been developed enormously in the recent years, enabling the analysis of the decision-making process to be more rigorous and robust, thus yielding highly reliable data modeling, especially in the case of structural equation modeling (SEM), which has been increasingly associated with the use of sociopsychological models in the last decade. It is also possible to state that the expansion of this technique has allowed researchers to increasingly study the behavioral aspects of adopting agricultural innovations.

For future investigations, the challenge remains to understand which factors have more weight in the success of the adoption and expansion of agricultural innovation, whether they are behavioral, institutional, economic, or ecological aspects, or even the characteristics of the innovation itself and how to relate all these aspects for a more complete analysis. The emergence of models emphasizing these areas in a balanced and contextualized way will bring advances in the perception of adoption. The extension of existing models is a good weapon that researchers and policymakers should explore, because it allows for a glimpse of other determinants of adoption not foreseen by the model's precursor.

This review demonstrated that, during the last decade, the number of studies on the adoption of agricultural innovations that employed these behavior-type models to understand farmers' decision process increased considerably in response to the various criticisms made regarding the lack of considering models with sociopsychological variables in the adoption process and its poor measurement. This aspect can be improved since most of these models allow for the addition of new constructs depending on the specificity of the study context and its objectives.

The results in aggregate form show a remarkable overlap of constructs in several sociopsychological models (Table 4), both in their original versions and in their extensions; the results indicate that the most frequently used models were TBP, UTAU, TAM, TRA, and MM. Constructs that were more repeated in the models were attitude, perceived usefulness, and perceived ease of use. The less frequently used constructs were efficacy, adoption opportunity, trust, awareness, environmental responsibility, and knowledge. The direct consequence of this overlap of constructs is a loss of explanatory power of the sociopsychological models. This enables us to conclude that, for a more complete analysis, able to capture all aspects of the decision-making process, an integrative approach [75] with economic models is necessary. These results are corroborated by Meijer et al. [1], who suggested an integrative approach to determine the intrinsic and extrinsic factors influencing farmers' adoption of sustainable practices.

In addition, our results evidenced that "knowledge" as a construct has barely been studied despite being relevant in the adoption decision. The literature review also showed that the application of models other than the TPB to agriculture has increased. However, the PBC and its constructs and extensions, as well as the TAM, the agent-based model, and the TRA, are the most used, indicating some confidence and comfort on the part of researchers in using these models. Even when these models are not used, the inclusion of constructs such as attitude, subjective norms, or PBC into the new sociopsychological models emerging and applied to agriculture is remarkable. This observation does not allow us to fully conclude that these models provide the best explanation of decision making on whether or not to adopt sustainable practices. However, it indicates that, when deciding which model to use, the researcher considers the problem of overlapping constructs that resulted from the robustness analysis in the studies reviewed.

**Table 4.** Sociopsychological models with overlapping determinants of adoption of sustainability agricultural innovations: results of vote-count methodology.

| Construct | Frequency of the Constructs | Sig. (+) | Sig.(−) | Models with Overlapped Constructs |
|---|---|---|---|---|
| Attitude | 28 | 28 | | TPB, UTAU, TAM, TRA, MM |
| PBC | 22 | 22 | | TPB, VBNT, TRA |
| Subjective norms | 21 | 21 | | TAM, UTAU, TPB, VBNT, TOE, PMT |
| Perceived usefulness | 18 | 18 | | TAM, UTAU, TPB, TTM, TOE, PMT |
| Perceived ease of use | 17 | 17 | | TAM, UTAUT, TOE |
| Perceived risk | 7 | | 7 | PMT, UTAUT, TPB, MOM |
| Intention | 5 | 5 | | TPB, TRA |
| Knowledge | 4 | 4 | | TAM, TBP, MM |
| Motivation | 4 | 4 | | MOA, MM, SCT |
| Relative advantage | 3 | 3 | | DOI |
| Believes | 2 | 2 | | PMT, MOM |
| Mitigation behaviour | 2 | 2 | | VBNT |
| Awarness | 2 | 2 | | MM, TRA |
| Environmental responsability | 2 | 2 | | VBNT |
| Triability | 2 | 2 | | DOI |
| Efficacy | 2 | 2 | | SCT |
| Trust | 1 | 1 | | UTAUT |
| Customer's readness to use | 1 | 1 | | TOE |
| Adoption opportunity | 1 | 1 | | MOA |

Note: TAM-Technology acceptance model; TRA-Theory of reasoned action; DOI-Diffusion of innovations; TPB-Theory of plannedbehaviour; VBNT-Values-believes norms theory; TTM-Transtheoretical model; SCT-Social cognitive theory; MOA-Motivation opportunity ability; TOE-Technology organisation environment; PMT-Protection motivation theory; MOM-Moral obligation model; MM-The multiplicity model; UTAU-Unified theory of acceptance and usage of technology.

## 5. Conclusions

The paper addressed a research gap in the literature consisting of a poor selection of the determinants included in the models explaining the adoption of sustainable agriculture innovations. The bibliometric review permitted the identification of the main trends in this topic, and we divided the analysis into two periods: from 2000 to 2009 and from 2010 to 2021. In the first period, the research patterns on innovation adoption were relatively stable, with an oscillating level of publications. This period was marked by the impact of previous studies that suggested including the interrelation of innovations, policies, and suggestions, as well as sociopsychological constructs, when modeling farmers' decision to adopt sustainable practices. The second period was marked by innovative ideas on how to evaluate the complementarity between scientific and local knowledge, as well as multidisciplinary cooperation among researchers to positively influence adoption. The results indicate a new dynamic regarding a broader use of sociopsychological factors in modeling adoption, facilitated by statistical techniques such as SEM, where the TPB, TAM, and TM highlighted constructs such as attitude, with PBC, SN, PU, and PEOU highlighting a positive and significant effect on the adoption of sustainability agriculture innovations, alongside the traditional determinants of the economic theory of UM. The results showed a remarkable overlap of constructs in several sociopsychological models, in both their original versions and their extensions (Table 4). The repetition of constructs suggests that the selection of the sociopsychological constructs leads to insufficient ability of the models to explain farmers' behavior to adopt sustainability agriculture innovations. There is a need for a better selection of the innovation determinants, which need to be adjusted to the study area contexts, for integrating sociopsychological factors with socioeconomic determinants. A better measurement of sociopsychological factors in the case of agriculture is also required.

On the other hand, the literature review showed that applying other models and including constructs such as attitude, subjective norms, or PBC into the new sociopsychological models that have emerged and been employed in agriculture can lead to remarkable

results. Despite recognizing the heterogeneity of the concept of adoption and specificity in terms of context, recent research has not yet provided evidence to fill this gap. Therefore, the emergence of models that are more specific to the contexts of adoption, as well as a combination of multi- and transdisciplinary models from economics, marketing, and sociopsychological models and their extensions, is increasingly recommended to explain the adoption of sustainable agricultural innovations. Sustainable agriculture was one of the most studied topics, and we noted a greater growth in the use of sociopsychological models, which may require researchers to adopt an integrative approach, thus building trans- and multidisciplinary models.

The novelty of the work was the systematization of the knowledge produced in the area of modeling the adoption of sustainability agriculture innovations, allowing for future researchers to obtain insights into the best-suited models, particularly regarding the selection of sociopsychological constructs that are influential in preventing the repetition of a poor selection.

As a theoretical implication, the effectiveness of sociopsychological models in the study of the adoption of agricultural innovations was confirmed through a literature review, and the attention was concentrated on constructs such as attitude, perceived usefulness, and perceived ease of use. An integrative approach with economic models, as well as the inclusion of relevant constructs such as efficacy, adoption opportunity, trust, awareness, environmental responsibility, and knowledge in future research, is important in the adoption of sustainable innovations. The research results have practical impact by promoting the adoption of sustainability agriculture innovations, as well as academic impact by raising researchers' attention to the need for expanding the dimensions represented by the models when analyzing the decision-making process. The study might incentivize other researchers to explore underestimated constructs.

This study's limitations relate to the fact that it was not possible to compare the most used models and the types of sustainable innovations, to assess whether different innovations are represented in the same way, by the same adoption or learning process. Another limitation of this study is that it was not possible to include more studies in the review. However, the study sample can be considered representative, which allowed us to answer the research question.

For future research, it will be relevant to understand if there are more adequate models to analyze adoption in the context of developed versus developing countries and their implications. This will require more cooperation among researchers from both regions to overcome the challenges in access to information.

**Author Contributions:** Conceptualization, J.R. and L.M.; methodology, J.R.; software, R.S.; validation, R.S. and C.M.; formal analysis, C.M.; investigation, J.R.; resources, J.R. and R.S.; data curation, R.S.; writing—original draft preparation, J.R.; writing—review and editing, L.M. and J.R.; visualization, C.M.; supervision, L.M. and C.M. All authors read and agreed to the published version of the manuscript.

**Funding:** This research was funded by the Portuguese Foundation for Science and Technology (FCT), the Ministry of Science, Technology, and Higher Education (MCTES), the European Social Fund (FSE) through NORTE 2020 (North Regional Operational Program 2014/2020), and the European Union (EU) through Grant 2020.07852.BD. The work of author Rui Silva was supported by national funds, through the FCT—Portuguese Foundation for Science and Technology under the project UIDB/04011/2022 and by NECE-UBI, Research Center for Business Sciences, under the project UIDB/04630/2022.

**Acknowledgments:** The authors would like express gratitude to CETRAD for all the support, as well as the Portuguese foundation for Science and Technology (FCT). The authors would also like to specially thank the reviewers for their constructive comments. Many thanks also go to the University Katyavala Bwila in Angola for the support.

**Conflicts of Interest:** The authors declare no conflict of interest.

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
