# Peer review of "Understanding Farmers’ Adoption of Sustainable Agriculture Innovations: A Systematic Literature Review"

_agronomy, doi:10.3390/agronomy12112879_

Round 1

Reviewer 1 Report

Dear Authors,

In my opinion, to be published, your paper requires substantial improvements. Please see my comments.

- The motivation for performing this study and the goals of the analysis, need to be presented in the introductory item, as well as the novel contributions for the reference literature. Now the elements of novelty and purpose of the article are not clear, what is done by the authors? This section should contain a justification of the need and relevance of the study (the size of this section is 1.5-2 pages);

- Theoretical background need to be presented. Many scientific schools investigated these variables, add authors, group them according to certain criteria (schools, countries, etc.), discuss what is not yet solved;

- There is room for improving the research protocol and providing a detailed step by step illustrated model with dependent and independent variables formulation and explanation. A clearer description on research model needs to be presented. Please separate research protocol including quantitative and qualitative methods explanation.

- A clearer description on article inclusion and exclusion criteria need to be presented. What is done to avoid bias during the entire process;

- How the decision-making process models were extracted from the articles?;

- Selection process of keywords need to be presented. No combined keywords that are relevant for the topic are included such as: “sustainable innovation”; “eco-innovation”; “ecological innovation” etc. keywords are not combined which lowers the probability to extract relevant publications;

- Please separate quantitative results from qualitative according to the changes made in methods section;

- One of the major contributions of such literature review studies is the future research agenda. Please provide separate section for future studies based on results. My advice is to group results according to certain criteria (schools, countries, etc.), discuss what is not yet solved and provide future research agenda;

- Although the authors present the added value of this paper, we consider that a highlight of the practical and academic utility of this synthesis should be better revealed.

- I would recommend emphasizing the conclusions of this study and key takeaways without repeating the information that was stated in the above sections;

- Highlight the limitations of the study;

- General conclusion is the article is pseudoscientific, the limited theoretical background and the use of inadequate methodological platform has led to false results and conclusions.

Reviewer 2 Report

The abstract is mostly mentioning the results of the study. It is recommended to add, in a few words, the motivation and the gap for this study.

 The methodology (bibliometric systematic literature review ) is well justified as a general approach.

 The search keywords should be justified, why these words not others?

 Using both WoS and Scopus to identify the dataset for the study gives richness, however a justification of this choice over other databases is still needed. Furthermore, R Bibliometrix 3.0 software requests a reference for its validity.

 Step 5 in the data collection process needs more clarifications in order to understand the publications selection process.

 The authors divided the dataset – dropping out the year 2022 - into 2 time periods: (2000-2009) and (2010-2021), where they mentioned: “The steep increase in publications in the last period must be understood as a result of the research agenda up-date due to food prices increasing in 2008, which once again placed agriculture on the agenda”, this argument needs a reference. Based on figure 2, the year 2015 could be a turning point for the increase of scientific production in relation to agriculture sustainability, and could be, randomly, explained due to the launching of the SDGs.

 Since this study is a systematic literature review, the future lines of investigation deserve more attention to provide more opportunities for future research.

 Most of the references in the manuscript don’t match their numbers in the Reference List, some of them even are not included. References should be revised.

Reviewer 3 Report

This is an important study, which explored the current research dynamics on models explaining farmers' decision-making processes on adopting sustainable agricultural innovations. Please find the following comments for further improvement.

Abstract

Line 22 on page 1 and line 50 on page 2, “which models have…”, the “w” of “which” should be capitalized.

Introduction

For the abbreviation first show up, it should be like “full name (abbreviation)”. Then in the following main text, you could directly use the abbreviation. Such as the “theory of reasoned action (TRA)” shows up twice in line 77 and line 88 on page 2, and “values-beliefs norms (VBN)” first shows up on page 6 line 338, but the abbreviation “VBN” first shows up on page 4 line 155. Please check every term of abbreviation in this manuscript and the similar issue, and fix them. Since this study involves a very large number of abbreviations, please pay attention to this issue.

Should a table for a list of abbreviations be added?

Where is the research gap? What are the main contributions of this study?

For references [1], [2], [3],[4], for the adoption of innovations in agriculture, it has been so many Review papers published, then the authors need to clarify the differences between this study and these existing reviews, and where the innovation of this study lies.

Methods

From line 122 to 126, the keywords for literature retrieval “"agricult* AND innovation AND adoption AND models OR theory OR adopt OR transtheoretical AND model OR bass- 123 like AND model OR sustainable OR diffusion OR goal-directed AND behaviour AND 124 attitude OR tpb OR task-technology AND fit AND model OR technology AND acceptance 125 AND model OR desire OR intention OR adoption AND behaviour”.

The search keyword string is so long, and the logic is not so clear. How do the keywords relate to others, and how the author did the search? Are so many search terms put into the search box all at once? These issues should be explained in detail.

Line 132 on page 3, why the reference [3] is cited here?

Lines 160-161 on page 4, the passage here is not a complete sentence.

Results and Discussion

For Figure 2, does this figure is drawn by authors or not? If not, is it exported from some kind of software? If so, the data source and the software should be reported in the note of the figure. What does the sign in the lower right corner of the blue-shaded graph mean? The fonts of the numbers and text in the picture are too small to see clearly (in both figure 2, figure 4, and figure 6). Additionally, the information in the title of the figure should be complete, but the title of the figure is too simple. Similar to the title of figures 4 and 5.

For Figures 3 and 5, it seems that it is derived from bibliometric software, which software? What bibliometric method was used? The results inside this figure were obtained by analyzing what data? These issues should also be explained and reported in the note of the figure.

In Figure 4, the color for the “Adoption”, “Technology” and “Technology Adoption”, the colors of the three lines are too similar to distinguish them.

Line 195, line 212, and line 243, it might not be appropriate to start a sentence with a Table/Figure reference.

The page numbers don't seem to be consecutive.

Line 273, the font of “Figure 5” is bigger than the others. Please maintain consistency.

Figure 6. the meaning of the different levels in the circle, the size of the area of the different colors, and the numbers in the figure should be explained.

Table 1 would be valuable for further research.

Table 2, please try not to let the table span pages.

Some words in table 3 (in the first column “Construct”) are not shown completely, please adjust.

References

Lines 126 to 127, “The data collection procedures were carried out in September 2022 and included papers from 2000 to 2022.” As the latest literature is updated to September 2022, however, why is there not a single paper from 2022 in the references?

It is necessary to learn more about the latest literature in 2022.

For references 13, 27, 30, 32, 34 and etc., the information on page numbers is lost.

Most information about the Year is in bolded font, but some are not.

Should the list of the 111 documents about the “Vote-count of models and constructs” be attached to the appendix or supplementary material?

Reviewer 4 Report

First of all, I would like to congratulate all the authors for a very interesting work. I think it is very important to have a good understanding on prior work conducted and bibliometric analysis is a great way to do this. However, I think the manuscript can be further improved to make it more interesting. My suggestions are as follows:

Abstract

Line 22: which should be written with a capital 'W'

Line 24-30: Sentence too long and should be broken into two sentences so that it is clear to the reader.

Introduction

Line 39: Would be helpful if the authors can list some examples of the sustainable agricultural innovations. This will be clearer for the reader I reckon. 

Line 46: Please rephrase the sentence. If possible, please avoid using 'how', 'what' as these are more suitable for questions than statement. 

Line 53: which should be with a capital 'W'

Line 76-79: Please rearrange the sentence to make it more concise and easier to comprehend

Line 84: evidence or evidences?

Line 85-86: 'concluded that including...' please rephrased this sentence

Line 91: identify or identified?

Line 92: show or showed

Line 94: Is 'This approach' referring to both TPM and UM. If yes, please correct the sentence accordingly. 

line 99: 'farmer level' is incorrect...do you mena farmers' level? 

Line 99-102: Please check and rephrase this sentence

Line 113-117: Please rephrased the sentence. Analysis or analyses? 

Line 117-119: Please remove this sentence. Would be interesting if the time frame of the publication selected is mentioned here instead.

Methods:

Line 123: Please include a bracket with OR eg adoption AND (models OR theory OR...) AND ...

Line 128: publication or publications?

Line 131: Please remove 'where'

Line 135: space missing between 'to' and 1052

Line 140-151: please check the indent and the format of the paragraph

Line 160: I find this sentence is rather abrupt and seems a little off

Results

Line 172: publication or publications?

Line 175: 'knowledge base regrading agriculture' sounds confusing to me. Please rephrase.

Line 176: Please replace 'last period'

Line 195: Please rephrase the sentence. There is a full stop after Figure 3

Line 212: Please avoid the word how in a statement 

Line 212-214: Please recheck the sentence

Line 216-222: Sentence too long. Perhaps you can break this down to two sentences instead. 

Line 243: The image should come after the first mentioned in the text, not the other way round. Please correct accordingly.

Line 250: adoption 'and' conservation agriculture? 

Line 273: Could you please provide a bit more info for Figure 5

Line 275: Please check the sentence

Line 280: Repetition

Line 298: argued? Please also check the sentence so that it written in a way that is clear to the reader

Line 300: Please avoid the word 'why'. Please rephrased accordingly.

Line 304: Just Table 1 in the bracket would suffice

Line 314: Please check the sentence

Line 325: I suggest the reference comes after the author's name

Line 360-363: Please rephrase the sentence

Line 371: Please remove the word 'how'

Line 395: Showed?

Line 396: 'The" should be with a capital T

Line 404: Please avoid the word 'who'

Line 412: "Subjective' with a capital 'S'?

Line 434: showed?

Conclusion
Line 483: Please remove 'aimed'. It should be the present study reviewed....

Line 490: Policy or policies?

Line 494: results indicated

Line 504-510: Please break the sentence up into 2 sentences. Too long and confusing

Line 517-523: Ditto comment above

References

Inconsistencies in the writing style as indicated in the manuscript. Please check the reference list carefully. Thank you.

Overall, the manuscript provides a good overview of the thought process involved in adopting a strategy for sustainable agricultural practices. The authors mention that all the models studied measure intentions rather than adoption/execution. Can you suggest how this can be improved so that actual adoption can be accurately measured?

The idea of comparing developed and developing countries in terms of adoption of sustainable innovations and the potential impact on their agricultural sector is very interesting. Perhaps the authors can explain the possible challenges in obtaining this information? Will this comparison be possible in the new future?

Round 2

Reviewer 1 Report

Despite the authors' claim that they made all corrections, there are comments that are not taken into account, namely.

·       - The motivation for performing this study and the goals of the analysis, need to be presented in the introductory item, as well as the novel contributions for the reference literature. Now the elements of novelty and purpose of the article are not clear, what is done by the authors? There is an exponential growth of articles that are investigating determinants of sustainable innovations, thus a need for such article and a gap of current state of knowledge need to be addressed.

·       - Theoretical background need to be presented. Many scientific schools investigated these variables, add authors, group them according to certain criteria (schools, countries, etc.), discuss what is not yet solved;

·       - There is room for improving the research protocol and providing a detailed step by step illustrated model with dependent and independent variables formulation and explanation. A clearer description on research model needs to be presented. Please separate research protocol including quantitative and qualitative methods explanation. Please address research methods and separate bibliometric analysis from SLR, hence these are two separate methods that follow separate research protocol.

·      -  A clearer description on article inclusion and exclusion criteria need to be presented. What is done to avoid bias during the entire process;

·       - Justification and selection process of keywords need to be presented – why these keywords?

·       - Please provide separate section for future studies based on results. My advice is to group results according to certain criteria (schools, countries, etc.), discuss what is not yet solved and provide future research agenda;

·       - Although the authors present the added value of this paper, we consider that a highlight of the practical and academic utility of this synthesis should be better revealed.

·       - Highlight the limitations of the study based on the limitations inherent in this type of research methods;

·       - With such heavy editing new mistakes and typos occurred.

·       - General conclusion is the article is pseudoscientific, the limited theoretical background and the use of inadequate methodological platform has led to false results and conclusions.

Reviewer 3 Report

The manuscript has been improved after revision. Please find the following comments for minor revisions.

1. The revised manuscript with all marks is hard to read. Please upload the clean version of the revised manuscript next time.

2. For “Figure 1 on page 8, should it be a “Table” rather than a “Figure”?

3. For “Figure 2” of section “4. Results and Discussion”, why the number of articles decreased in 2021 compared to 2020? This should be explained.

4. Figure 5 Co-citation networks, why the “feder g 1985” showed up twice, with one big note and another smaller note?

5. The introduction and literature review could be improved by focusing more on the research questions and contents rather than the authors (who).